# Clinical implication and potential function of ARHGEF6 in acute myeloid leukemia: An *in vitro* study

**Kang Li[1], Haiquan Wang[1], Chaofan Yang[1], Chaojun Li[1]\*, Bin Xue[2]\*, Jiankui Zhou[3]\***

**1** Medical School of Nanjing University, Nanjing, Jiangsu, China, **2** Core Laboratory, Sir Run Run Hospital, Nanjing Medical University, Nanjing, China, **3** Precise Genome Engineering Center, School of Life Sciences, Guangzhou University, Guangzhou, Guangdong, China

\* zhoujiankui@126.com (JZ); xuebin@njmu.edu.cn (BX); licj@nju.edu.cn (CL)

## Abstract

The roles of Rho GTPases in various types of cancer have been extensively studied, but the research of Rho guanine nucleotide exchange factors (GEFs) in cancer is not comprehensive. Rho guanine nucleotide exchange factor 6 (ARHGEF6) is an important member of the Rho GEFs family involved in cytoskeletal rearrangement, and it has not been investigated in acute myeloid leukemia (AML). Our research showed that the expression of ARHGEF6 was mainly higher in AML cell lines, meanwhile, was highest in the samples from patients with AML compared to other cancer types. High ARHGEF6 expression in AML was associated with a good prognosis. ARHGEF6[low] cases showed significantly higher overall survival (OS) after autologous or allogeneic HSCT (auto/allo-HSCT). High expression of ARHGEF6 downregulates the negative regulation of myeloid differentiation process and upregulates G protein-coupled receptor signaling pathway-related processes, among which HOXA9, HOXB6, and TRH have significant differential expression and prognostic impact in AML. Therefore, ARHGEF6 can become a prognostic marker in AML; ARHGEF6[low] patients can gain from auto/allo-HSCT.

## Introduction

AML is characterized by accumulation of immature cells resulting from uncontrolled proliferation of myeloid progenitor cells, thus impairing myeloid differentiation and ultimately decrease the percentage of normal blood cells. In the United States, 75% of AML patients are over 65 years old, elder patients are refractory and prone to relapse, the recurrence rate can reach 10%-40% even in younger patients [1, 2]. Recently, several studies have shown that AML patients with some genes (such as *ARHGAP9* and *BCL2*) abnormal expression can benefit from auto/allo-HSCT [3, 4]. Therefore, optimal therapeutic strategy based on validated prognostic markers is important for AML patients.

Rho GTPases belong to the Ras GTPase family, and are activated at cellular membranes by Rho GEFs [5, 6]. Activated Rho GTPases participate in various biological processes, for instance, vesicle transport and cytoskeletal rearrangement [7, 8]. In cancer, Rho GTPases are

**Data Availability Statement:** All relevant data are within the paper and its Supporting Information files.

**Funding:** This work was supported by Guangdong Basic and Applied Basic Research Foundation

(2020A1515110581 to J. Zhou), Science and Technology Program of Guangzhou (202201020209 to J. Zhou), start-up funds from Guangzhou University to J. Zhou, Chinese National Science Foundation (32071145 and 31771572 to B. Xue), the Nature Science Foundation of Jiangsu Province (BK20191356 to B. Xue). There was no additional external funding received for this study.

**Competing interests:** The authors have declared that no competing interests exist.

thought to be correlated with tumor development and poor prognoses [9]. And in hematopoiesis, Rho GTPases are concerned with various processes such as cell proliferation, differentiation, migration, and self-renewal [10–14]. Rho GTPases have significant impacts on both cancer and the hematopoietic system.

Rho GEFs are considered as prospective targets for cancer treatment, because of their functions to promote the GTP-bound state formation of Rho GTPases [15]. Although some Rho GEFs are overexpressed in cancer tissues and exhibit poor prognoses [16–18], the functional and clinical significance of most Rho GEFs remain undefined. ARHGEF6 (αPIX/Cool2) is identified as a GEF of Rac1/Cdc42, binds with PARVB and CAPNS1, participates in the regulation of cytoskeletal rearrangement, including cell adhesion and migration [19, 20]. In gliomas, ARHGEF6 overexpression correlates with tumor grading [21]. However, ARHGEF6 signaling has an essential role in apoptosis induction in chlorambucil-resistant ovarian carcinoma [22]. These instances demonstrate that the function of ARHGEF6 is distinctively according to the specific type of cancer.

Although ARHGEF6 expression has been confirmed in platelets [23], there have been no research about the expression and function of ARHGEF6 in AML. Our research showed the ARHGEF6 expression and its correlation with clinicopathological characteristics in AML. We further assessed the prognostic significance of ARHGEF6 and discussed its impact on the choice of AML treatment.

## Materials and methods

### ARHGEF6 expression analysis in cell lines

ARHGEF6 expression, on the mRNA level, was analyzed in the Human Protein Atlas (HPA) database (https://www.proteinatlas.org) [24]. On the entry of ARHGEF6, the "Cell Line" section based on genome-wide RNA expression was chosen, and transcriptomic data were sorted using the "Organ" parameter. ARHGEF6 expression, on the protein level, was analyzed in the Cancer Cell Line Encyclopedia (CCLE) database (https://portals.broadinstitute.org/ccle) [25]. On the entry of ARHGEF6 in the "CCLE data" section, proteomics was selected, and protein data were downloaded followed by plotting with the ggplot2 package in R 3.3.5.

### Pan-cancer analysis of ARHGEF6 expression

ARHGEF6 expression pattern in pan-cancer was conducted using the UALCAN database (http://ualcan.path.uab.edu/) [26]. Briefly, "Pan-cancer view" was chosen on the entry of ARHGEF6 and the expression of ARHGEF6 across TCGA tumors was exhibited. Pan-cancer analysis of ARHGEF6 expression pattern was also conducted by employing the Gene Expression Profiling Interactive Analysis (GEPIA) database (http://gepia.cancer-pku.cn) [27], showing the expression profile of ARHGEF6 across various tumors with paired normal tissues. The expression results in the AML are also obtained in the GEPIA database through the "Expression DIY" tool.

### ARHGEF6 expression in AML with different karyotypes

The Gene Expression Omnibus (GEO) database (https://www.ncbi.nlm.nih.gov/geo) was used to get ARHGEF6 expression patterns in AML with different karyotypes. ARHGEF6 expression data in GSE14468 were retrieved from the GEO2R online software based on ARHGEF6 ID "209539_at". GSE14468 has 526 AML patients in total. The raw counts were used to generate the data in GraphPad Prism 8.0.

## AML clinical data analysis

LinkedOmics database (http://www.linkedomics.orglogin.php) was used to analyze the relationship between ARHGEF6 expression and OS in AML patients [28]. Briefly, "TCGA-LAML" cancer type, "RNA-seq" data type, "ARHGEF6" gene name, and "clinical" target data type were chosen.

This study comprised a cohort of 173 AML patients having ARHGEF6 expression data from TCGA (https://cancergenome.nih.gov/ and http://www.cbioportal.org/) [29]. For consolidation treatment, 73 patients received auto/allo-HSCT, while the remaining 100 patients got just chemotherapy. Based on the mRNA level of ARHGEF6, these patients were separated into two groups (ARHGEF6$^{low}$ and ARHGEF6$^{high}$) (S1 File). Table 1 summarizes the key clinical and laboratory characteristics of cases with different ARHGEF6 expressions.

## Transcriptome analysis and functional annotation

RNA-seq data of AML patients were downloaded from TCGA database and normalized using the quantile normalization procedure. Differential expressed genes (DEGs) between ARHGEF6$^{high}$ and ARHGEF6$^{low}$ groups were identified by t-test in the limma package. And if the adjust P value < 0.05 and log(FoldChange) (logFC) > 1, we considered the RNAs to be differentially expressed.

Enrichment analysis of Gene ontology (GO) and Kyoto Encyclopedia of Genes and Genomes (KEGG) were performed by ClusterProfiler package in R software and we considered the terms with P value less than 0.05 to be significant.

## Prognostic validation of DEGs

We validated the prognosis of DEGs between the ARHGEF6 high and low expression groups by using the survival analysis panel in GEPIA database. Briefly, we selected the "survival plot" under survival analysis, entered the gene name, selected the "LAML" dataset.

## Statistical analysis

IBM SPSS 26 was used to do statistical analyses of the data. Categorical variables were compared by chi-square test and Fisher's exact tests. Because the number of samples in each group was fewer than 5000, the Shapiro–Wilk test was employed to determine if the values in each group were normally distributed for the comparison of continuous variables. A two-sample t-test or the Mann–Whitney U test was employed, depending on the values were normal/abnormal distributed, respectively. Except for the LinkedOmics database, the Log-rank in GraphPad Prism 8.0 was used to examine the prognostic impact of ARHGEF6 expression and different treatments on Disease-free survival (DFS) and OS.

# Results

## ARHGEF6 overexpressed in AML

To determine ARHGEF6 expression in AML cells, we analyzed RNA-seq and proteomics data in the HPA and CCLE databases, respectively. In the HPA database, ARHGEF6 mRNA expression levels in myeloid cell lines such as HEL, HL60, HMC-1, and U937 were higher than in lymphoid cell lines. Meanwhile, ARHGEF6 mRNA was almost non-existent in the lung, reproductive system, skin, and other tissues (S1A Fig). Furthermore, in the CELL database, ARHGEF6 had the highest protein level in AML cell lines (S1B Fig).

We next examined the mRNA expression of ARHGEF6 in various human tumor samples using the UALCAN and GEPIA databases. ARHGEF6 mRNA expression level was the highest

**Table 1. Correlations between ARHGEF6 expression and clinicopathological characteristics in AML from the TCGA cohort.**

| Patient characteristics | ARHGEF6 expression | | |
| --- | --- | --- | --- |
| | Low (n = 87) | High (n = 86) | p |
| Sex, male/female | 47/40 | 45/41 | 0.823 |
| Median age, years (range) | 61 (18–88) | 55.5 (21–77) | 0.103 |
| Median BM blasts, % (range) | 73 (30–99) | 72 (32–100) | 0.709 |
| Median WBC,(range) $\times 10^9$/L | 14.9 (0.7–297.4) | 19.15 (0.4–223.8) | 0.773 |
| Median PB blasts, % (range) | 29 (0–98) | 41 (0–97) | 0.162 |
| WHO classifications AML with certain genetic abnormalities | 12 | 26 | 0.009 |
| RUNX1-RUNX1T1 | 1 | 6 | 0.064 |
| CBFB-MYH11 | 3 | 7 | 0.211 |
| PML-RARA | 5 | 11 | 0.110 |
| MLLT3-KMT2A | 1 | 0 | 1 |
| RBM15-MKL1 | 1 | 0 | 1 |
| BCR-ABL1 | 1 | 2 | 0.621 |
| AML-MRC | 30 | 26 | 0.55 |
| t-AML | NA | NA | NA |
| NOS | 45 | 33 | 0.078 |
| M0 | 0 | 3 | 0.121 |
| M1 | 17 | 8 | 0.056 |
| M2 | 9 | 12 | 0.467 |
| M4 | 9 | 6 | 0.431 |
| M5 | 9 | 3 | 0.132 |
| M6 | 0 | 0 | 0 |
| M7 | 1 | 0 | 1 |
| No data | 0 | 1 | 0.497 |
| Risk level Good | 9 | 23 | 0.005 |
| Intermediate | 55 | 46 | 0.194 |
| Poor | 22 | 15 | 0.208 |
| NA | 1 | 2 | 0.621 |

*n* number of patients, *WHO* World Health Organization

*AML-MRC* AML with myelodysplasia-related changes

*t-AML* Therapy-related AML, *NOS* not otherwise specified

*BM-blast* bone marrow blast

*WBC* white blood cell

*PB-blast* peripheral blood blast

across all kinds of cancers (Fig 1A and 1B). Moreover, in the GEPIA database, a significantly higher ARHGEF6 expression level was found in AML compared to normal tissues (Fig 1C). Then microarray data (GSE14468) was utilized to find out whether ARHGEF6 expression was associated with major recurrent chromosomal translocations in the GEO database. The result showed that AML patients with t(8;21) had the highest ARHGEF6 expression compared with other karyotypes (Fig 1D).

## The association of ARHGEF6 with clinicopathological characteristics of AML patients

Table 1 summed up the clinical characteristics of patients according to the clinical data from the TCGA database. The WHO classification and risk stratification were significantly different

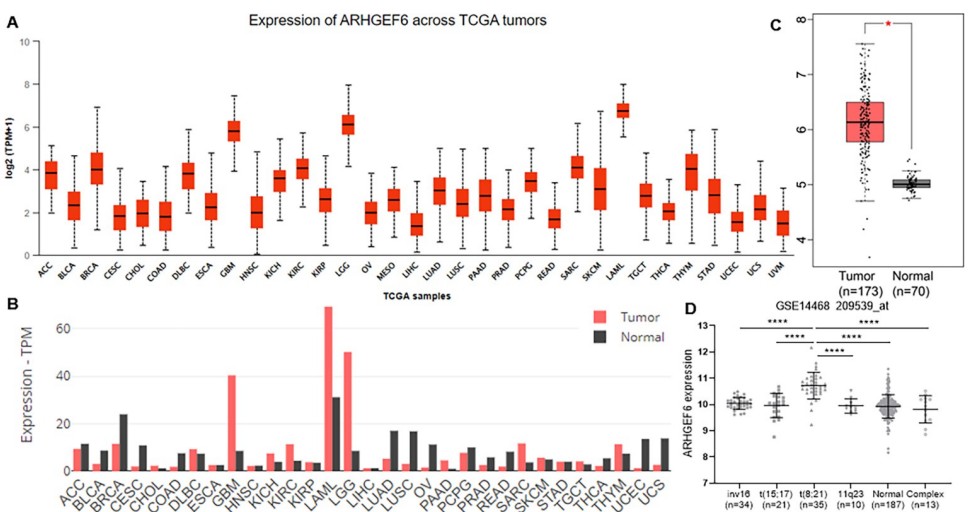

**Fig 1. ARHGEF6 expression in AML.** (A) ARHGEF6 expression in pan-cancer, result from UALCAN. (B) The expression of ARHGEF6 in different cancerous and normal tissues using the GEPIA. Bar height refers to the value of the median expression. (C) ARHGEF6 expression between AML and normal samples, result was obtained by GEPIA analysis. *, p < 0.05. (D) ARHGEF6 expression in AML among various karyotypes, according to GSE14468 in the GEO database. ****, p < 0.0001.

in patients with different ARHGEF6 expressions. In the WHO classification distribution, high ARHGEF6 expression was significantly associated with genetic abnormalities. In addition, patients with high ARHGEF6 expression tended to have good prognoses. Meanwhile, there was no significant difference in age, sex, BM blasts, WBC, or PB cells between the ARHGEF6[low] and ARHGEF6[high] cases (Table 1).

## Prognostic values of ARHGEF6 in AML

To investigate the relations between ARHGEF6 expression and prognosis in AML patients, we assessed the effect of ARHGEF6 expression on OS by employing the LinkedOmics database. The result indicated that low ARHGEF6 expression in AML was associated with poor OS (Fig 2A). Meanwhile, the survival data from the TCGA database was analyzed, as a result, ARHGEF6 overexpression was significantly associated with higher OS (Fig 2B). The DFS was higher in the ARHGEF6[high] cases compared with the ARHGEF6[low] cases, although it did not reach statistical significance (Fig 2C).

In cytogenetically normal AML (CN-AML), patients with different ARHGEF6 expressions had no significant difference in OS or DFS (Fig 2D and 2E). Furthermore, no significant differences in OS or DFS was detected between the ARHGEF6[low] and ARHGEF6[high] groups with either chemotherapy or chemotherapy plus auto/allo-HSCT treatment (Fig 2F–2I). Overall, low ARHGEF6 expression in AML exhibited a poor prognosis. However, neither chemotherapy alone nor chemotherapy combined with auto/allo-HSCT improved the prognosis of ARHGEF6[low] patients. Finally, in ARHGEF6[low] groups, patient survival with or without (WOW) auto/allo-HSCT were analyzed. The result showed that ARHGEF6[low] patients who received auto/allo-HSCT had significantly improved OS, but not DFS, than patients undergoing only chemotherapy (Fig 2J and 2K).

## Functional annotation of DEGs between ARHEGF6[low] and ARHGEF6[high] group

To find the function of ARHGEF6 in AML, we analyzed the RNA-seq data of the ARHGEF6 low and high expression groups. We obtained a total of 504 DEGs, of which 163 genes were

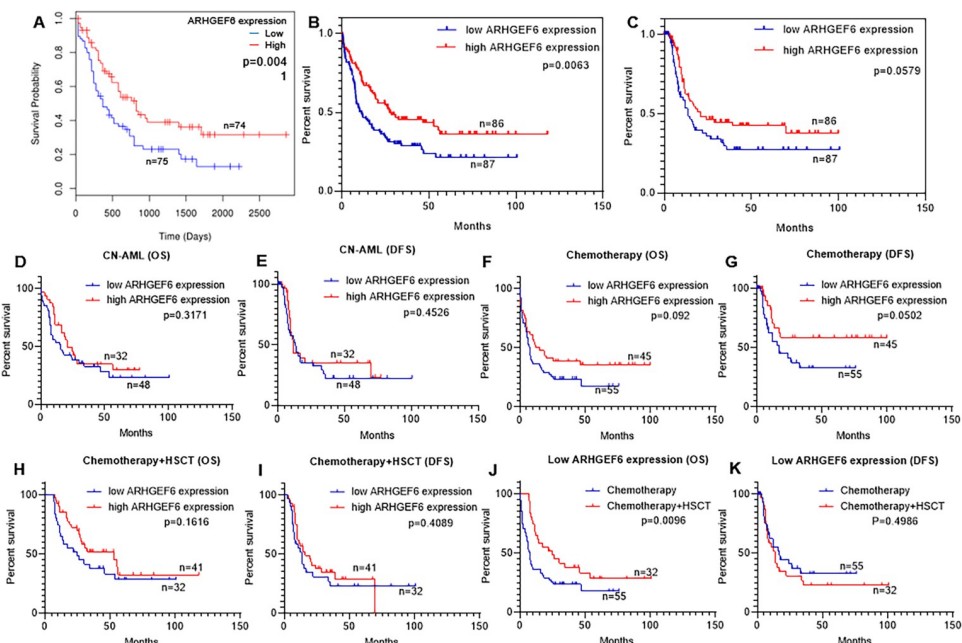

**Fig 2. Correlation between AML patient survival and ARHGEF6 expression with different factors.** (A) The prognostic value between ARHGEF6$^{low}$ and ARHGEF6$^{high}$ group using the LinkedOmics database. (B) OS and (C) DFS in AML patients with different ARHGEF6 expressions. (D) OS and (E) DFS in CN-AML patients with different ARHGEF6 expressions. (F) OS and (G) DFS of ARHGEF6$^{low}$ and ARHGEF6$^{high}$ patients treated with chemotherapy. (H) OS and (I) DFS of ARHGEF6$^{low}$ and ARHGEF6$^{high}$ patients undergoing chemotherapy + auto/allo-HSCT. (J) OS and (K) DFS of patients treated WOW auto/allo-HSCT in the ARHGEF6$^{low}$ group.

significantly up-regulated and 341 genes were significantly down-regulated (S2 File), and we marked the top 9 significantly highly expressed and top 10 significantly lowly expressed genes, respectively (Fig 3A).

GO analysis annotated Molecular Function (MF), Cellular Component (CC), and Biological Process (BP) that were significantly up- and down-regulated with increasing ARHGEF6 expression levels, respectively (Fig 3B and 3C, S3 File). The results showed that the significantly up-regulated genes were mainly involved in the classical regulatory pathways of GEFs, such as G protein-coupled receptor signaling pathway, synapse assembly, and vesicle transport. The significantly down-regulated genes were mainly involved in the differentiation and development of the embryonic skeletal system and the negative regulation of myeloid differentiation.

KEGG pathway annotation showed that the significantly up-regulated pathways mainly involved circadian entrainment, renin secretion, vascular smooth muscle contraction, transcriptional misregulation in cancer, etc.; the significantly down-regulated pathways included osteoclast differentiation, B cell receptor signaling pathway, cytokine receptor interaction, signaling pathways regulating pluripotency of stem cells, hippo and wnt signaling pathway, etc (Fig 3D, S4 File).

## Prognosis validation of DEGs

We validated the prognostic profile of AML for the top 10 genes that were significantly up- and down-regulated, and the results showed that high expression of the top 10 genes that were significantly up-regulated in patients were all associated with higher OS, among which, *TRH* significantly high expressed in AML samples compared to normal samples (Fig 3E), and high

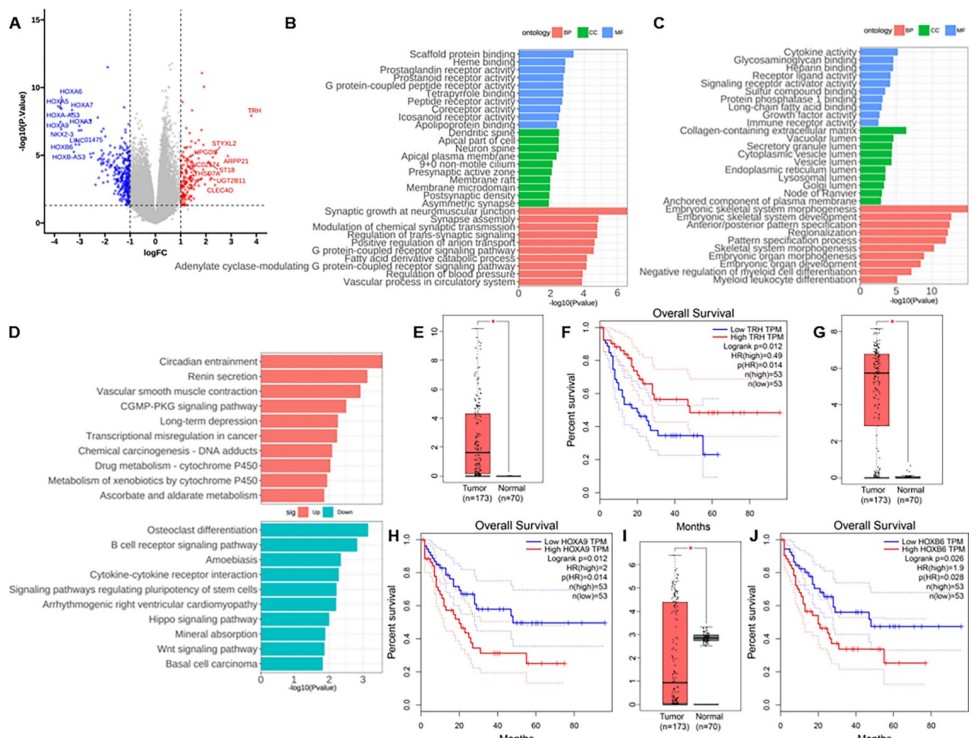

**Fig 3. Functional annotation and prognosis validation of differentially expressed genes between ARHEGF6$^{low}$ and ARHGEF6$^{high}$ group.** (A) Volcano plot of DEGs between the ARHGEF6$^{low}$ and ARHGEF6$^{high}$ groups. (B) GO analysis of significantly upregulated genes. (C) GO analysis of significantly downregulated genes. (D) KEGG analysis of DEGs. (E) and (F) TRH expression levels in AML and normal tissues, and the effect on OS in AML patients, result from GEPIA. *, p < 0.05. (G) and (H) HOXA9 expression levels in AML and normal tissues, and the effect on OS in AML patients, result from GEPIA. *, p < 0.05. (I) and (J) HOXB6 expression levels in AML and normal tissues, and the effect on OS in AML patients, result from GEPIA. *, p < 0.05.

expression of *TRH* in AML patients was a good prognostic factor that was significantly associated with higher OS (Fig 3F); High expression of the significantly down-regulated genes were shown to be associated with lower OS in AML patients, among which, *HOXA9* and *HOXB6* having significantly different expression profiles in AML samples compared to normal samples (Fig 3G and 3I, respectively), and high expression in AML were poor prognostic factors (Fig 3H and 3J).

## Discussion

Rho GEFs activate Rho GTPases by exchanging bound GDP with GTP [30]. Numerous studies have pointed out that abnormal expression of Rho GEFs has been found in human cancers [31, 32]. Rho GEFs expression is distinct in different cancer types. For example, DOCK2 is upregulated in follicular lymphoma and downregulated in NSCLC (non-small cell lung cancer) [33, 34]. Tiam1 is downregulated in colorectal cancer and highly expressed in various cancers such as gastric, laryngeal squamous cell carcinoma, and ovarian cancers [35–37]. In the current study, ARHGEF6 was highly expressed among AML tissues and cell lines, especially in t(8;21) AML patients with relatively good prognoses.

Some studies on Rho GEFs have shown that a large part of Rho GEFs are correlated with poor prognosis in various tumors [38–40]. For example, ABR, PREX1, DOCK2, and DOCK4 showed poor prognosis in NSCLC [31]. But, in AML, the high expression group of ARHGEF6

had a good prognosis. Thus, Rho GEFs play different roles in different cancers. On the other hand, ARHGAP9, which belongs to the Rho GAP family, inactivates GTPases and oppositely function to that of Rho GEFs. ARHGAP9 is an adverse prognostic factor for AML OS [3]. Thus, Rho GEFs and Rho GAPs may play other roles in cancers.

Despite the significantly lower expression of ARHGEF6 in CN-AML, no relationship between ARHGEF6 expression and CN-AML prognosis has been found. Furthermore, several researchers found that some genes, for example, ARHGAP9 and BCL2 were correlated with the prognosis of auto/allo-HSCT and/or chemotherapy in AML patients [3, 4]. Our findings showed that auto/allo-HSCT significantly improves prognosis in patients with low ARHGEF6 expression. However, in the high ARHGEF6 expression group, patients treated with chemotherapy + auto/allo-HSCT had a significant decrease in DFS, although there was no significant change in OS. These results suggest that patients with low ARHGEF6 expression could benefit from autologous/allogeneic HSCT, but autologous/allogeneic HSCT is not recommended for patients with high ARHGEF6 expression because of the significantly higher tendency to relapse and progression.

Interestingly, after analysis of patient RNA-seq data, we found that high expression of ARHGEF6 was associated with the downregulation of several HOX gene family members. the HOX gene family plays a crucial regulatory role in animal development [41], and in hematopoiesis, the HOX gene family is involved in regulating the differentiation and developmental process of hematopoietic stem cells to different cell types [42], and the disturbances in their expression levels are associated with the development of AML [43]. Previous studies have identified that HOXA9 overexpressed in AML and is a poor prognostic factor [44], consistent with our results. In vivo, HOXB6 promotes the development of AML by promoting the proliferation of hematopoietic stem cells and myeloid precursor cells while inhibiting the production of erythroid and lymphocytes [45]. On the other hand, the significantly upregulated gene TRH was found to be associated with a good prognosis in a study of t(8;21) acute myeloid leukemia [46]. However, there are no studies about the role of TRH in AML.

Overall, we first reported that ARHGEF6 overexpressed in AML cell lines, tissues, and the t (8; 21) patients. Elevated ARHGEF6 expression was significantly correlated with a favorable prognosis in AML. A combination of auto/allo-HSCT and chemotherapy, instead of only chemotherapy, can improve poor prognosis related to low ARHGEF6 expression. High expression of ARHGEF6 downregulated the expression level of the poor-prognosis HOXA9 and HOXB6, while increasing the expression of the good-prognosis TRH, thus improving the OS of patients.

## Supporting information

**S1 Fig. ARHGEF6 expression in cell lines.** (A) The mRNA levels of ARHGEF6 in human cell lines using the HPA. (B) The relative protein levels of ARHGEF6 in human cell lines using the CCLE. Labels of x-axis were sorted from large to small according to the median expression of protein.
(TIF)

**S1 File. ARHGEF6<sup>low</sup> and ARHGEF6<sup>high</sup> group information.** Patients were divided into ARHGEF6<sup>low</sup> group (n = 87) and ARHGEF6<sup>high</sup> group (n = 86) according to the mRNA expression level of ARHGEF6.
(TXT)

**S2 File. DEGs list.** DEGs between ARHGEF6<sup>low</sup> and ARHGEF6<sup>high</sup> groups.
(XLSX)

**S3 File. DEGs GO.** GO analysis of the DEGs in S2 File.
(XLSX)

**S4 File. DEGs KEGG.** KEGG analysis of the DEGs in S2 File.
(XLSX)

## Acknowledgments

We thank Guangzhou University for supporting the study.

## Author Contributions

**Conceptualization:** Kang Li.

**Data curation:** Haiquan Wang, Chaofan Yang.

**Formal analysis:** Chaofan Yang.

**Funding acquisition:** Bin Xue, Jiankui Zhou.

**Investigation:** Kang Li.

**Project administration:** Bin Xue, Jiankui Zhou.

**Resources:** Chaojun Li, Jiankui Zhou.

**Supervision:** Chaojun Li.

**Visualization:** Haiquan Wang.

**Writing – original draft:** Kang Li.

**Writing – review & editing:** Jiankui Zhou.

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
