## [Decision Letter · Decision Letter 0]

12 Jan 2023

PONE-D-22-12620Clinical implication and potential function of ARHGEF6 in acute myeloid leukemiaPLOS ONE

Dear Dr. Zhou,

Thank you for submitting your manuscript to PLOS ONE. After careful consideration, we feel that it has merit but does not fully meet PLOS ONE’s publication criteria as it currently stands. Therefore, we invite you to submit a revised version of the manuscript that addresses the points raised during the review process.

We have received the opinions of expert reviewer and we invite you to submit a revised version of the manuscript. Please consider and address each of the comments raised by the reviewers.  

We look forward to receiving your revised manuscript.

Kind regards,

Senthilnathan Palaniyandi, Ph.D

Academic Editor

PLOS ONE

Journal Requirements:

“This work was supported by Guangdong Basic and Applied Basic Research Foundation (2020A1515110581 to J. Zhou), start-up funds from Guangzhou University to J. Zhou, Chinese National Science Foundation (32071145 and 31771572 to B. Xue), the Nature Science Foundation of Jiangsu Province (BK20191356 to B. Xue).”

Reviewers' comments:

Reviewer's Responses to Questions

**Comments to the Author**

1. Is the manuscript technically sound, and do the data support the conclusions?

Reviewer #1: Partly

Reviewer #2: Yes

2. Has the statistical analysis been performed appropriately and rigorously? 

Reviewer #1: Yes

Reviewer #2: Yes

3. Have the authors made all data underlying the findings in their manuscript fully available?

Reviewer #1: Yes

Reviewer #2: Yes

4. Is the manuscript presented in an intelligible fashion and written in standard English?

Reviewer #1: Yes

Reviewer #2: Yes

5. Review Comments to the Author

Reviewer #1: This article is bioinformatic and it just collected published data and then analyzed them. Most of data derived from cell lines, the experiment result of cell line are artificial data not is not real data. we discuses on AML it meaning patients life, by statically analysis we can reach to use one drug is good or not. Any Patient is unique.

Reviewer #2: Thank you for the opportunity to review the manuscript titled ‘Clinical implication and potential function of ARHGEF6 in acute myeloid leukemia’.

The authors have highlighted an important finding of high ARHGEF6 expression in patients with AML being associated with a favorable prognosis and low expression being associated with a poor prognosis. The authors have also highlighted the improvement of outcomes by using hematopoietic stem cell transplantation in patients with ARHGEF6 low expression. Additionally, the role of the HOX gene family and TRH has been highlighted which is an important prognostic finding. The paper is well written, informative and I believe will be valuable to readers given its scientific merit.

The only point I would like to make is that FAB classification of AML is now obsolete and it would be interesting to see the trends of ARHGEF6 expression with the new classification.

6. PLOS authors have the option to publish the peer review history of their article (what does this mean?). If published, this will include your full peer review and any attached files.

Reviewer #1: No

Reviewer #2: **Yes: **Supriya Gupta, MD

---

## [Author Response · Author response to Decision Letter 0]

10 Feb 2023

Dear Dr. Senthilnathan Palaniyandi, 

Thank you for your letter dated January 12. We were pleased to know that our work was rated as potentially acceptable for publication in PLOS ONE, subject to adequate revision. We thank the reviewers for the time and effort that they have put into reviewing the previous version of the manuscript. Their suggestions have enabled us to improve our work. Based on the instructions provided in your letter, we uploaded the file of the revised manuscript. Accordingly, we have uploaded a copy of the original manuscript with all the changes highlighted by using the track changes mode in MS Word. Appended to this letter is our point-by-point response to the comments raised by the reviewers. The comments are reproduced and our responses are given directly afterward. We would like also to thank you for allowing us to resubmit a revised copy of the manuscript.

We hope that the revised manuscript is accepted for publication in PLOS ONE. 

Sincerely, 

Jiankui Zhou

Encl. Responses to the comments from Reviewers #1 and #2.

Reply to Reviewer #1

Dear Reviewer,

Thank you very much for your time involved in reviewing the manuscript; your responsible attitude toward patients is admirable.

Comments:

“This article is bioinformatic and it just collected published data and then analyzed them. Most of data derived from cell lines, the experiment result of cell line are artificial data not real data. we discuss on AML it meaning patients life, by statically analysis we can reach to use one drug is good or not. Any Patient is unique.”

We also appreciate your clear and detailed feedback and hope that the explanation has fully addressed all of your concerns.

In this work, we focused on the expression characteristics, clinical prognosis, and potential function of the Rho guanine nucleotide exchange factor6 (ARHGEF6) in acute myeloid leukemia (AML). Through analysis of multiple databases (2 datasets on cell line; 4 datasets on RNA levels of patients; 2 datasets on patients’ clinical records), we found that:

(1) As a member of Rho guanine nucleotide exchange factors (GEFs), ARHGEF6 expression in AML has not been reported. By analyzing several databases, we found that ARHGEF6 is highly expressed in tissues and cell lines of AML. Pan-cancer analysis of ARHGEF6 showed its highest expression in AML. In the analysis of several common AML recurrence-associated karyotypes, we found that ARHGEF6 expression was significantly higher in t(8;21) compared to other karyotypes.

(2) Previous studies have generally concluded that Rho GEFs play a role in cancer promotion because they activate Rho GTPases. But a large part of Rho GEFs has not been sufficiently studied in various cancer types. Our analysis of the TCGA database and LinkedOmics database revealed that overall survival was significantly higher in patients with high ARHGEF6 expression, suggesting that overexpression of ARHGEF6 is associated with a good prognosis in AML. Besides, low ARHGEF6 expression patients can benefit from auto/allo-HSCT. These results can guide clinical treatment.

(3) ARHGEF6 downregulates the expression level of HOX family genes which inhibit hematopoietic stem cell differentiation, meanwhile increasing the expression of TRH, a favorable prognostic factor in AML. These results somewhat corroborate our previous findings and reveal the role of ARHGEF6 in AML.

Although we used data from cell lines, this part of the data only presented in FigS1, and all data presented in the text are from AML patients. 

Providing treatment plans to patients need to be personalized, and we should be responsible for the patient's life. Basic research is a long process from bench to bedside. Research starts from the cellular level, then move up to the animal, and finally is validated with precious clinical patient samples. For humans, the number of genes to be studied is massive, and screen for some genes with key roles in a limited time is crucial. Bioinformatic analysis can accelerate this process and guide subsequent laboratory and clinical studies.

We would like to take this opportunity to thank you for all your time involved and this great opportunity for us to improve the manuscript. We hope you will find this revised version satisfactory.

Sincerely,

Jiankui Zhou

-----End of Reply to Reviewer #1------

Reply to Reviewer #2

Dear Dr. Supriya Gupta,

Thank you very much for your time in reviewing the manuscript and your encouraging comments on the merits.

Comments:

“The authors have highlighted an important finding of high ARHGEF6 expression in patients with AML being associated with a favorable prognosis and low expression being associated with a poor prognosis. The authors have also highlighted the improvement of outcomes by using hematopoietic stem cell transplantation in patients with ARHGEF6 low expression. Additionally, the role of the HOX gene family and TRH has been highlighted which is an important prognostic finding. The paper is well written, informative and I believe will be valuable to readers given its scientific merit.”

We also appreciate your clear and detailed feedback and hope that the explanation has fully addressed all of your concerns. In the remainder of this letter, we discuss each of your comments individually along with our corresponding responses. 

Comment1:

“The only point I would like to make is that FAB classification of AML is now obsolete and it would be interesting to see the trends of ARHGEF6 expression with the new classification.”

Response1:

Thanks for your great suggestion on improving the accessibility of our manuscript. We have re-analyzed the data using the WHO classification published in 2016. The relevant contents are provided below as a screen dump for your quick reference.

The WHO classification and risk stratification were significantly different in patients with different ARHGEF6 expressions. In the WHO classification distribution, high ARHGEF6 expression was significantly associated with genetic abnormalities.

We would like to take this opportunity to thank you for all your time involved and this great opportunity for us to improve the manuscript. We hope you will find this revised version satisfactory.

Sincerely,

Jiankui Zhou

-----End of Reply to Reviewer #2------

---

## [Decision Letter · Decision Letter 1]

8 Mar 2023

PONE-D-22-12620R1Clinical implication and potential function of ARHGEF6 in acute myeloid leukemiaPLOS ONE

Dear Dr. Zhou,

Thank you for submitting your manuscript to PLOS ONE. After careful consideration, we feel that it has merit but does not fully meet PLOS ONE’s publication criteria as it currently stands. Therefore, we invite you to submit a revised version of the manuscript that addresses the points raised during the review process.

We have now received comments from the referee of your manuscript, we invite you to submit a revised version of the manuscript. Would you be able to consider adding in vitro in the title?  

We look forward to receiving your revised manuscript.

Kind regards,

Senthilnathan Palaniyandi, Ph.D

Academic Editor

PLOS ONE

Journal Requirements:

Reviewers' comments:

Reviewer's Responses to Questions

**Comments to the Author**

1. If the authors have adequately addressed your comments raised in a previous round of review and you feel that this manuscript is now acceptable for publication, you may indicate that here to bypass the “Comments to the Author” section, enter your conflict of interest statement in the “Confidential to Editor” section, and submit your "Accept" recommendation.

Reviewer #1: All comments have been addressed

2. Is the manuscript technically sound, and do the data support the conclusions?

Reviewer #1: Yes

3. Has the statistical analysis been performed appropriately and rigorously? 

Reviewer #1: I Don't Know

4. Have the authors made all data underlying the findings in their manuscript fully available?

Reviewer #1: Yes

5. Is the manuscript presented in an intelligible fashion and written in standard English?

Reviewer #1: Yes

6. Review Comments to the Author

Reviewer #1: I read the answers of correspond auteur, it described very well. as I suggest in last review. Same experiment is needed on primary AML cells derived from patients in two groups: M3 and non M3 groups.

Or change the title : added in vitro results

7. PLOS authors have the option to publish the peer review history of their article (what does this mean?). If published, this will include your full peer review and any attached files.

Reviewer #1: No

---

## [Author Response · Author response to Decision Letter 1]

20 Mar 2023

Dear Dr. Senthilnathan Palaniyandi, 

Thank you for your letter dated March 9. We are pleased to know that our work was rated as potentially acceptable for publication in PLOS ONE, subject to adequate revision. We thank the reviewer for the time and effort in reviewing the previous version of the manuscript. The suggestions have enabled us to improve our work. Based on the instructions provided in your letter, we revised the article title and uploaded the file of the revised manuscript. Accordingly, we have uploaded a copy of the original manuscript with all the changes highlighted by using the track changes mode in MS Word. Appended to this letter is our response to the comment raised by the reviewer. We would like also to thank you for allowing us to resubmit a revised copy of the manuscript.

We hope that the revised manuscript is accepted for publication in PLOS ONE. 

Sincerely, 

Zhou Jiankui

Encl. Responses to the comments from Reviewer #1.

Reply to Reviewer #1

Dear Reviewer,

Thank you very much for your time involved in reviewing the manuscript. Your suggestions are of great help to us.

Comments:

“I read the answers of correspond auteur, it described very well. as I suggest in last review. Same experiment is needed on primary AML cells derived from patients in two groups: M3 and non M3 groups. Or change the title : added in vitro results”

We also appreciate your clear and detailed feedback. In this revised version we have changed the title to “Clinical implication and potential function of ARHGEF6 in acute myeloid leukemia: an in vitro study”.

We would like to take this opportunity to thank you for all your time involved and this great opportunity for us to improve the manuscript. We hope you will find this revised version satisfactory.

Sincerely,

Zhou Jiankui

-----End of Reply to Reviewer #1------

---

## [Editor Report · Decision Letter 2]

22 Mar 2023

Clinical implication and potential function of ARHGEF6 in acute myeloid leukemia：an in vitro study

PONE-D-22-12620R2

Dear Dr. Zhou,

We’re pleased to inform you that your manuscript has been judged scientifically suitable for publication and will be formally accepted for publication once it meets all outstanding technical requirements.

Kind regards,

Senthilnathan Palaniyandi, Ph.D

Academic Editor

PLOS ONE
---

## [Editor Report · Acceptance letter]

29 Mar 2023

PONE-D-22-12620R2 

Clinical implication and potential function of ARHGEF6 in acute myeloid leukemia: an *in vitro* study 

Dear Dr. Zhou:

I'm pleased to inform you that your manuscript has been deemed suitable for publication in PLOS ONE. Congratulations! Your manuscript is now with our production department. 

Kind regards, 

on behalf of

Dr. Senthilnathan Palaniyandi 

Academic Editor

PLOS ONE